# Novel Antimicrobial Peptide “Octoprohibitin” against Multidrug Resistant *Acinetobacter baumannii*

**DOI:** 10.3390/ph15080928

**Published:** 2022-07-27

**Authors:** E. H. T. Thulshan Jayathilaka, Dinusha C. Rajapaksha, Chamilani Nikapitiya, Joeun Lee, Mahanama De Zoysa, Ilson Whang

**Affiliations:** 1College of Veterinary Medicine and Research Institute of Veterinary Medicine, Chungnam National University, Yuseong-gu, Daejeon 34134, Korea; thimira.thulshan@o.cnu.ac.kr (E.H.T.T.J.); dinusharajapaksha@o.cnu.ac.kr (D.C.R.); chamilani14@cnu.ac.kr (C.N.); 2National Marine Biodiversity Institute of Korea (MABIK), 75, Jangsan-ro, 101beon-gil, Janghang-eup 33662, Korea; le408@mabik.re.kr

**Keywords:** *Acinetobacter baumannii*, antimicrobial peptide, biofilm inhibition, biofilm eradication, multidrug resistance, octoprohibitin, zebrafish

## Abstract

Octoprohibitin is a synthetic antimicrobial peptide (AMP), derived from the prohibitin-2 gene of *Octopus minor*. It showed substantial activity against multidrug resistant (MDR) *Acinetobacter baumannii* with a minimum inhibitory concentration (MIC) and minimum bactericidal concentration (MBC) of 200 and 400 µg/mL, respectively. Time-kill kinetics and bacterial viability assays confirmed the concentration-dependent antibacterial activity of octoprohibitin against *A. baumannii*. The morphology and ultrastructure of *A. baumannii* were altered by treatment with octoprohibitin at the MIC and MBC levels. Furthermore, propidium iodide-fluorescein diacetate (PI-FDA) staining and 2′,7′-dichlorodihydrofluorescein diacetate (H_2_DCFDA) staining of octoprohibitin-treated *A. baumannii* revealed membrane permeability alterations and reactive oxygen species (ROS) generation, respectively. Agarose gel retardation results confirmed the DNA-binding ability of octoprohibitin to the genomic DNA of *A. baumannii*. Furthermore, octoprohibitin showed concentration-dependent inhibition of biofilm formation and eradication. The minimum biofilm inhibition concentration (MBIC) and minimum biofilm eradication concentration (MBEC) of octoprohibitin were 1000 and 1460 µg/mL, respectively. Octoprohibitin produced no significant cytotoxicity up to 800 µg/mL, and no hemolysis was observed up to 400 µg/mL. Furthermore, in vivo analysis in an *A. baumannii*-infected zebrafish model confirmed the effective bactericidal activity of octoprohibitin with higher cumulative survival percent (46.6%) and fewer pathological signs. Histological analysis showed reduced alterations in the gut, kidney, and gill tissues in the octoprohibitin-treated group compared with those in the phosphate-buffered saline (PBS)-treated group. In conclusion, our results suggest that octoprohibitin is a potential antibacterial and antibiofilm agent against MDR *A. baumannii.*

## 1. Introduction

*Acinetobacter baumannii* is an opportunistic pathogen, associated with most hospital-derived Acinetobacter infections [1]. *A. baumannii* infection is frequent in immunocompromised patients with skin and soft tissue diseases, surgical sites, bloodstream infections, and urinary tract infections [2]. Aerobic, non-fermentative, Gram-negative, non-motile bacillus, *A. baumannii,* harbors multiple virulence factors to survive in a wide range of adverse conditions and persists on surfaces for a long time in an inactive form. The possession of an intrinsic efflux pump, genetic adaptation, biofilm formation, and the presence of several beta-lactamase inhibitors are the main defense mechanisms of *A. baumannii* for the development of antibiotic resistance [3]. In recent studies, *A. baumannii* was found to be resistant to multiple antibiotics, including carbapenem, which is the last line of drugs for controlling it [4]. At present, combination antibiotic therapy is the final solution used in the medical fraternity for controlling multidrug resistant (MDR) *A. baumannii,* which makes treatment procedures more complex and has more incidences of side effects [5]. The development of alternatives to antibiotics is the only solution for overcoming the challenges of MDR *A. baumannii* [6].

Tackling antibiotic resistance has become a major challenge in the healthcare system, and thus far, multiple compounds, such as probiotics, prebiotics, organic acids, antimicrobial peptides (AMPs), nanomaterials, and bacteriophages, have been tested as possible candidates [7,8,9]. Most of AMPs are natural antimicrobial compounds synthesized by organisms from archaea to mammals as a group of compounds involved in innate immunity. AMPs play a key role in defending the host against pathogenic bacteria, fungi, protozoa, and viruses [10]. AMPs are diverse within and across species, sharing common features consisting of 10 to 50 amino acid (AA) residues, water solubility, and high hydrophobicity with a positive charge (majority of AMPs). Depending on their secondary structure, AMPs can be divided into α-helices, β-sheets, and extenders [11]. AMPs possess multiple modes of action, from initial membrane disruptions to DNA and RNA damage, induction of reactive oxygen species (ROS) generation, and denaturation of proteins and enzymes [12]. Since AMPs acquire multiple modes of action and are not constrained to a single pathogen species, the possibility of the development of AMP resistance by the pathogen is at the lowest possible level [13]. Moreover, the majority of newly discovered and synthesized AMPs have shown extremely low or no toxicity towards humans and other tested animals [14]. Considering the characteristics and functional features of AMPs with low toxicity, AMPs have potential for the next generation of antimicrobials.

Marine invertebrates are deficient in adaptive immunity and primarily depend on innate immunity, which consists of physical and chemical compounds, including AMPs [15]. *Octopus minor* is considered a rich source of novel AMPs, and we synthesized octominin and octopromycin in previous studies [16,17]. Octominin, which is based on defense protein-3, showed potent bactericidal and fungicidal activities against *A. baumannii* and *Candida albicans,* respectively, whereas octopromycin, which is based on proline-rich 5 protein, showed high antibacterial activity against *A. baumannii* [16,17,18]. Prohibitin is a highly conserved class of proteins that inhibits cellular proliferation and has other roles in nuclear signaling, mitochondrial integrity, cell division, and cellular membrane metabolism [19]. We found the prohibitin-2 gene in the *O. minor* transcriptome, which was selected for the development of octoprohibitin as a novel AMP. Initially, we tested the antibacterial activity of octoprohibitin against *A. baumannii* and its detailed modes of action, including morphological and structural changes of the surface, membrane permeability alterations, ROS, and DNA binding ability. Furthermore, we tested octoprohibitin as an antibiofilm agent by inhibition of biofilm formation and biofilm eradication assays. Ultimately, an *A. baumannii*-infected zebrafish model was used to determine in vivo efficacy of Octoprohibitin for controlling the infection.

## 2. Results

### 2.1. Designing, Synthesis, and Characterization of Octoprohibitin

The AA sequence of *O. minor* prohibitin-2 cDNA sequence was selected as a template to synthesize octoprohibitin. Octoprohibitin exhibited characteristic features of AMPs with a net positive charge of +10, a high number of positively charged residues (arginine and lysine), and a hydrophobic ratio of 38% with two hydrophobic AAs on the surface. Furthermore, octoprohibitin has a predicted aliphatic index of 63.46, a grand average hydropathicity value (GRAVY) of −1.146, and a molecular weight of 3333.14 Da. The predicted secondary structure of the alpha-helix (helical wheel projection diagram) and tertiary (three-dimensional) structure of octoprohibitin are shown in Figure 1. The HPLC analysis results indicated that the peak for octoprohibitin was obtained at 14.98 min (Appendix A).

### 2.2. Antibacterial Activity of Octoprohibitin against A. baumannii

Since octoprohibitin showed relatively lower antimicrobial activity against *A. baumannii* at pH 7.4, we used the broth microdilution method at pH 7, 8, and 9 to determine the optimum pH level and found that octoprohibitin showed efficient antibacterial activity with increasing pH (Appendix A). We selected Luria-Bertani broth (LB broth; Invitrogen, MA, USA) at pH 8 as the culture medium for further experiments because the optimum pH value for *A. baumannii* growth is 5–8 [20]. To confirm the antibacterial activity of octoprohibitin against *A. baumannii,* minimum inhibitory concentration (MIC) and minimum bactericidal concentration (MBC) were determined as 200 and 400 µg/mL, respectively, using the broth microdilution method and agar plating method. The calculated MBC/MIC ratio was 2.0. The tested positive control (chloramphenicol; 100 µg/mL) had MIC and MBC values of 50 and 100 µg/mL, respectively, against *A. baumannii.*

To investigate the antibacterial activity of octoprohibitin, time-kill kinetic and bacterial viability assays were conducted at different concentrations (0–400 µg/mL). Time-kill kinetic data showed a growth rate reduction at higher octoprohibitin treatment concentrations from 100 µg/mL, while total bacterial growth inhibition was observed at 300 µg/mL and above (Figure 2A).

The 3-(4,5-dimethylthiazol-2-yl)-2,5-diphenyltetrazolium bromide (MTT) assay confirmed the reduced bacterial viability following octoprohibitin treatment in a concentration-dependent manner. At lower concentrations, *A. baumannii* exhibited regular growth (Figure 2B). Significant bacterial viability reduction was observed at 50 µg/mL, with a viability of 81.33% (*p ≤* 0.05), and the inhibition continued to increase at higher concentrations. At the highest tested concentration (400 µg/mL), octoprohibitin demonstrated 36.83% bacterial viability, whereas chloramphenicol (100 µg/mL) showed 9.30% viability.

### 2.3. Morphological and Ultrastructural Changes in A. baumannii with Octoprohibitin Treatment

Field-emission scanning electron microscopy (FE-SEM) was conducted to examine the effect of octoprohibitin on the morphology and ultrastructure of *A. baumannii.* Micrographs of FE-SEM analysis revealed considerable damage to the *A. baumannii* cell surface at octoprohibitin MIC (200 µg/mL) and MBC (400 µg/mL) treatment, while bacteria in the negative control showed normal cell surface and shape (Figure 3). Furthermore, octoprohibitin-treated MBC showed prominent and more severe damage with open holes and cell shrinkage compared to that of the MIC-treated group; the damage was relatively identical to that of the positive control, chloramphenicol (100 µg/mL)-treated *A. baumannii* cells.

### 2.4. Membrane Permeability Alteration of A. baumannii with Octoprohibitin Treatment

With the confirmation of bacterial surface morphology alterations, a propidium iodide (PI) uptake assay was conducted to determine membrane permeability alteration in *A. baumannii* with octoprohibitin treatment. Fluorescence intensity was measured by flow cytometry (FCM), and micrographs were captured by confocal microscopy (CFMC). PI can only cross the cell membrane when permeability is altered, bind to DNA, and produce red fluorescence [21]. As expected, only the negative control (1 × PBS treated) showed red fluorescence (0.9%) compared to that of the blank (Figure 4A). Octoprohibitin treatment at MIC (200 µg/mL) and MBC (400 µg/mL) resulted in a high number of PI-stained cells with intensities of 14,689 and 18,557, respectively. Notably, chloramphenicol (100 µg/mL)-treated bacterial cells showed a fluorescence level of 14351. FCM analysis data were further confirmed using the micrographs of CFMC analysis with fluorescein diacetate (FDA) staining. FDA is permeable to live cells, and viable cells convert FDA into the green fluorescent metabolite fluorescein [22]. All the bacterial cells in the negative control showed green fluorescence (FDA-stained). Octoprohibitin at MIC, MBC and the positive control groups had prominent red fluorescein (PI-stained) cells, while a slight amount of green fluorescence cells were observed in the MIC-treated group (Figure 4B). These data suggest that octoprohibitin induces alterations in bacterial membrane permeability in *A. baumannii* in a concentration-dependent manner.

### 2.5. Induction of ROS Generation with Octoprohibitin Treatment in A. baumannii

To determine ROS generation following octoprohibitin treatment, FCM and CFMC analyses were conducted on octoprohibitin-treated *A. baumannii* with 2′,7′-dichlorodihydrofluorescein diacetate (H_2_DCFDA) staining. Flow cytometry analysis of *A. baumannii* treated at MIC (200 µg/mL) and MBC (400 µg/mL) showed higher green fluorescence levels with 4.99- and 8.51-fold compared to that of the negative control, respectively, and the chloramphenicol (100 µg/mL)-treated group only had 4.46-fold (Figure 5A). Moreover, CFMC analysis micrographs showed a high number of green fluorescent cells in the octoprohibitin-treated groups, while the MBC group had a relatively higher number of green fluorescent cells compared to the MIC-treated group (Figure 5B). The negative control group had no green fluorescent cells, while the highest number of green fluorescent cells was observed in the positive control group. These FCM and CFMC data confirmed octoprohibitin-induced ROS generation in *A. baumannii.*

### 2.6. DNA Binding Ability of Octoprohibitin with Genomic DNA of A. baumannii

Agarose gel retardation assay was conducted on the genomic DNA of *A. baumannii* treated with octoprohibitin to determine the DNA binding ability of octoprohibitin. Samples of genomic DNA treated with low concentrations of octoprohibitin did not exhibit significant deviation in degradation from the untreated sample (Figure 6A). However, when the treatment concentration was increased, the intensity of the DNA band decreased in a concentration-dependent manner. The lowest visible band was observed at an octoprohibitin: DNA ratio of 5:1. The DNA-binding ability of octoprohibitin was confirmed at higher ratios of octoprohibitin (7.5:1) treatments, with inhibition of genomic DNA migration and possible DNA digestion with smear appearance at those treatment levels (Figure 6B).

### 2.7. Biofilm Formation Inhibition and Eradication Effect of Octoprohibitin on A. baumannii

To quantify the remaining biofilm after octoprohibitin treatment, crystal violet (CV) staining was used in biofilm formation and biofilm eradication assays. Biofilm formation was inhibited in a concentration-dependent manner, with significant (*p ≤* 0.05) inhibition observed at 25 µg/mL (21.58%) and above (Figure 7A). The highest biofilm formation inhibition was observed at the highest tested concentration of 800 µg/mL (91.55%). The positive control chloramphenicol (100 µg/mL) treatment group showed an 87.53% inhibition of biofilm formation. The calculated minimum biofilm inhibitory concentration (MBIC) of octoprohibitin was 1000 µg/mL. The biofilm eradication assay showed a similar pattern of concentration-dependent biofilm eradication (Figure 7B). At the highest tested concentration (800 µg/mL), a significant (*p ≤* 0.05) biofilm eradication percentage (92.11%) was observed, whereas the positive control, chloramphenicol (100 µg/mL), had 92.34% eradication. The calculated minimum biofilm eradication concentration (MBEC) for octoprohibitin was 1460 µg/mL.

### 2.8. Hematotoxicity and Cytotoxicity of Octoprohibitin

Hemolysis assays on mouse red blood cells (RBCs) and cell viability assays on human embryonic kidney 293T (HEK293T) cells were conducted to determine the toxicity of octoprohibitin. Almost zero hemolysis was observed at octoprohibitin concentrations up to 400 µg/mL (Figure 8A). Detectable hemolysis was observed at 600 µg/mL of octoprohibitin with 26.89% of hemolysis and at 800 µg/mL, octoprohibitin showed 92.35% of hemolysis compared to that of the 100% of hemolysis in Triton x-100-treated group. In the cell viability test, a slight reduction in cell viability was observed with increasing concentrations; however, it was not significant up to 800 µg/mL. At octoprohibitin 1000 µg/mL, a significant (*p ≤* 0.05) reduction in bacterial viability (54.61%) was observed (Figure 8B).

### 2.9. In Vivo Effectiveness of Octoprohibitin in A. baumannii-Infected Zebrafish

To determine the antibacterial activity of octoprohibitin *in vivo*, adult zebrafish were challenged with *A. baumannii* and treated with octoprohibitin and tested for cumulative survival percentage (CS%), pathological signs, and histological analysis. Zebrafish that were infected and treated with PBS had the highest mortality, and the CS% at 48 h post treatment (hpt) was 13.33%. Fish that were challenged and the octoprohibitin-treated group had significantly (*p ≤* 0.05) higher CS (46.66%) than those in the PBS-treated group (Figure 9A). No mortality was observed in the negative control group throughout the experiment.

*A. baumannii*-infected and PBS-treated zebrafish showed pathological signs of red coloration in the abdomen area and gill area, while those signs were relatively lower in the octoprohibitin-treated group (Figure 9B). No pathological signs were observed in the negative group.

Histological analysis showed a significant deviation in the tissue structure of the PBS-treated zebrafish group compared with that of the negative control in the gut, kidney, and gill samples (Figure 9C). Specifically, the PBS-treated group showed hyperplasia in goblet cells and erythrocyte infiltration, with epithelial cell damage in the gut tissue sample. A relatively low number of goblet cells was observed in the octoprohibitin-treated gut sample, with the absence of erythrocyte infiltration and damage to the surface epithelial cells. Kidney tissues of the PBS-treated samples showed shrinkage in the glomerulus and infiltration of proximal and distal tubules with a low density of cells due to possible tissue necrosis. The octoprohibitin-treated sample showed minimal alterations from the negative control, with a regular tissue structure in the kidney. Clubbing and shortening of the secondary lamella and thickening of the primary lamella epithelium owing to hyperplasia were observed in the gill tissue samples of the PBS-treated zebrafish group. Tissue deformation was at a minimum level in octoprohibitin-treated gill tissue samples, and only slight thickening of the primary lamella epithelial cell layer was observed compared to that of the negative control.

## 3. Discussion

AMPs may be a one of major solutions for the control of MDR pathogens such as *A. baumannii,* since they contain multiple modes of action, are effective against a wide range of pathogenic species, are able to improve host immunity, and have low toxicity [23]. Marine invertebrates are rich in several AMPs because they rely mainly on innate immunity. In several previous studies, we studied *O. minor* for its AMPs and synthesized octominin and octopromycin, which showed potent antimicrobial activities against both fungi and bacteria [16,17]. In the present study, the AA sequence of the prohibitin-2 cDNA sequence was selected as a template for designing and synthesizing a novel AMP named octoprohibitin. The *A. baumannii* strain used in this study was screened for antibiotic sensitivity and was found to be resistant to multiple antibiotics in our previous study [24]. Initially, octoprohibitin showed slight antimicrobial activity against *A. baumannii* at a pH of 7.4. Some AMPs exhibit pH-dependent antimicrobial activity. Esculentin-2EM, an alpha-helical peptide isolated from *Glandirana emeljanovi*, showed a higher rate of antimicrobial activity against Gram-positive and Gram-negative bacteria at pH 8 than that at pH 6 [24]. We examined the optimum pH for the highest antibacterial activity of octoprohibitin and found more efficient bactericidal activity at pH 8 and 9. Since the optimum pH range for *A. baumannii* growth is 5–8, we selected pH 8 for *A. baumannii* culturing for further experiments with octoprohibitin treatments [25]. Potent antimicrobial activity of octoprohibitin was detected at MIC and MBC values of 200 and 400 µg/mL, respectively. The bactericidal or bacteriostatic activity of the antimicrobial agent is determined by the MBC/MIC ratio, and a compound is considered bactericidal if the ratio is below 4.0, and bacteriostatic if the ratio is higher than 4.0 [26]. Octoprohibitin is considered bactericidal rather than bacteriostatic because of its MBC/MIC ratio of 2.0. Moreover, a time-kill kinetic assay and bacterial viability test verified concentration-dependent antibacterial activity, similar to our previous finding with octominin and octopromycin against *A. baumannii* [17,18].

Cationic AMPs mainly act by disrupting the bacterial membrane. The positive charge of AMPs facilitates efficient binding to the negatively charged bacterial membrane and completes the activity through different models, such as forming pores and ion channels to induce overflow of bacterial content (barrel stave model) or disintegrating the cell membrane by changing the surface tension (carpet model) [27]. Since octoprohibitin possesses a +10 charge and amphiphilic nature, the main predicted mode of action against *A. baumannii* is morphological and structural damage to the bacterial surface and membrane permeability alterations. Micrographs of FE-SEM analysis confirmed that the surface morphology of the bacteria was altered by the formation of pores on the membrane and significant damage to the surface by octoprohibitin treatment. In addition, Mwangi et al. suggested that cathelicidin-based AMP ZY4 induces permeabilization of the *A. baumannii* membrane with a similar pattern of damage by examining the cell surface using FE-SEM [28]. According to the toroidal-pore model, cationic AMPs aggregate inside the cell membrane and alter the phospholipid monolayer of bacteria to produce nanoscale pores, which facilitates permeability alterations [29]. A PI uptake assay of octoprohibitin-treated *A. baumannii* confirmed a significant amount of membrane permeability alterations similar to the activity of octominin to induce membrane perturbation in *A. baumannii* [18]. Morphological alterations with pore formation and membrane permeability changes together might result in the leakage of the intracellular content of *A. baumannii* to the outside and cause cellular apoptosis or necrosis, ultimately causing bacterial death.

Several studies have shown that AMPs can penetrate the bacterial cell cytoplasm and induce further bactericidal activities by interrupting biochemical processes inside the cell [30]. Since octoprohibitin induces alterations in membrane surface morphology and permeability, we hypothesized that octoprohibitin could translocate into the cytoplasm and induce further antimicrobial actions. A study on LL-37, a cationic AMP, induced superoxide and peroxide production in *Escherichia coli* under aerobic conditions [31]. Increased ROS inside bacterial cells induces damage to proteins, lipids, and nucleotides, and this negative impact on bacteria is tolerable up to a certain level due to the activation of antioxidative stress response genes. Once an antimicrobial agent is capable of producing lethal stress with a significant amount of ROS, which is not borne by defense mechanisms, bacterial death is self-driven [32]. The latest findings suggest that generated ROS can cause eventual cell death by surging the self-amplification of ROS, even after the removal of the antimicrobial agent [33]. Flow cytometry and confocal microscopy analysis confirmed that octoprohibitin could increase the level of ROS inside *A. baumannii,* which could be a potential antimicrobial action that has not been well discussed previously with other AMPs in detail. However, further experiments are essential to determine the exact mechanism underlying ROS formation by octoprohibitin in *A. baumannii*.

Since we expected octoprohibitin translocation into *A. baumannii,* we predicted its cationic charge to facilitate possible binding to negatively charged genomic DNA for further expansion of the antimicrobial activity. Several AMPs depend on their antimicrobial activity by binding to bacterial DNA. BF2-A and BF2-C, which are buforin II-based AMPs isolated from Asian toads, can inhibit nucleic acid metabolism [34]. The ostrich-derived β-defensins ostricacin-1 (Osp-1) and ostricacin-2 (Osp-2) show antimicrobial activity by inhibiting protein synthesis and enzymatic activity as a result of their binding affinity to DNA [35]. Microcin-based B17 AMP can bind to DNA and block DNA replication by inhibiting the gyrase-DNA complex [36]. The gel retardation assay is the initial step in investigating the DNA-binding ability of an AMP. Since octoprohibitin showed concentration-dependent DNA binding and possible genomic DNA degradation in *A. baumannii,* it may contain further DNA-based antimicrobial activities. Further investigations are essential to determine the exact mechanism by which octoprohibitin binds to and disrupts the genomic DNA of *A. baumannii.*

*A. baumannii* has a remarkable capacity to form biofilms on a wide range of surfaces, from abiotic stainless steel to host epithelial cells, and it has been identified as one of the major factors for antibiotic resistance [37]. Antibiofilm AMPs can act adversely at different stages of biofilm formation through various mechanisms, including downregulation of quorum sensing, killing of early-stage biofilm, and inhibition of adhesion [38]. Octoprohibitin might also have generated similar actions to inhibit the formation of *A. baumannii* biofilms with an MBIC of 1000 µg/mL. Nevertheless, eradication of pre-formed biofilms by antimicrobials is considered challenging because biofilms consist of various defense mechanisms. To date, inadequate diffusion of antibiotics, efflux mechanisms, enzyme-facilitated deactivation, heterogenic function, slow development rate or persistent cells, and biofilm phenotype adaptive mechanisms have been identified [39,40]. We hypothesized that octoprohibitin could evade these defense mechanisms because it produced a potential disruption of the pre-formed *A. baumannii* biofilm in a concentration-dependent manner. Although octoprohibitin had significant antibiofilm effects in CV staining, further experiments are essential to define the exact inhibition and eradication mechanism of *A. baumannii* and its biofilm.

AMPs are considered to possess cell selectivity by differentiating the microbial membrane structure, and act only against pathogenic microorganisms without generating toxicity against host cells [41]. Several AMPs have been tested in vitro for their efficiency in antimicrobial activity, but their application in vivo is limited because of their toxicity to eukaryotic cells, susceptibility to proteolytic degradation, and the development of allergies [42,43]. Since octoprohibitin is positively charged and hydrophobic, we projected its selectivity towards bacterial cells and lowered its interaction with eukaryotic cells for the least toxic effects. Since the hemolysis concentration (600 µg/mL) and HEK293T viability reduction concentration (800 µg/mL) were higher than the MIC (200 µg/mL) and MBC (400 µg/mL), we confirmed the safety of octoprohibitin for application in zebrafish to determine its effectiveness in vivo against *A. baumannii* infection.

In our previous study, we developed a zebrafish model of *A. baumannii* infection and successfully tested it with octopromycin [17]. Because we could confirm the low toxicity of octoprohibitin in vitro, the zebrafish model was used to observe how octoprohibitin treatment controls *A. baumannii* infection in vivo. As expected, octoprohibitin demonstrated its antimicrobial activity in zebrafish against *A. baumannii* infection significantly (*p <* 0.05), showing high CS% compared to that of the control group. Since visible pathological signs of hemorrhage were observed in untreated fish in the gill and abdomen area, we expanded our study to determine alterations in tissue levels by histopathology analysis. Although *A. baumannii* was not well tested in the zebrafish model previously, we identified a significant level of tissue alterations in the gut, kidney, and gills. Generally, these tissues show necrosis, reduced cell volume, and possible hemorrhages. In particular, hyperplasia of goblet cells in the gut tissue and hyperplasia and clubbing of secondary lamella in the gill confirmed the inflammatory conditions caused by *A. baumannii* in the PBS-treated control group. However, the low level of tissue damage in the octoprohibitin-treated group was confirmed by the survival data and suggested a high level of control zebrafish from *A. baumannii* infection. IDR-1, which is a derivative of bovine bactenecin, does not possess direct antimicrobial properties in vitro but was found to be protective in several mouse models infected with Gram-negative and Gram-positive pathogenic bacteria [44]. Likewise, some AMPs exhibit immunomodulatory actions in the host, such as altered chemokine and cytokine release, leukocyte activation, mast cell degranulation, anti-endotoxin generation, and chemotaxis strengthening immunity [45]. As octoprohibitin was able to control *A. baumannii* infection in zebrafish, we can conclude that protection is either by directly killing the bacteria or indirectly modulating the immunity of the host, and/or by both actions. Further experiments are essential to determine the exact mode of action of octoprohibitin in controlling *A. baumannii* infection in vivo.

Even though we have tested octoprohibitin antibacterial activity against Gram-negative *A. baumannii*, owing to its vast mode of action, such as membrane permeabilization, ROS generation, DNA binding, and antibiofilm activities, octoprohibitin may produce antibacterial activities against other Gram-negative and Gram-positive bacteria, antiviral activities, and antifungal activities. Additionally, owing to the ROS generation properties of octoprohibitin, it may be developed as an anticancer drug. However, further studies on octoprohibitin are essential for elaborating its applications in other fields.

## 4. Materials and Methods

### 4.1. Design, Synthesis, and Characterization of Octoprohibitin

The transcriptome database of *O. minor* was screened for AMPs, and the prohibitin-2 cDNA sequence was considered a candidate for designing a novel AMP. The prohibitin-2 cDNA sequence was submitted to the National Center for Biotechnology Information (NCBI) (https://www.ncbi.nlm.nih.gov/, accessed on 7 January 2022) database with accession number MW939430. To design octoprohibitin, the C-terminal region of the prohibitin-2 AA sequence was selected as a template. The novel AMP, octoprohibitin (WRVCKQSEKVILNPFWKKKKKKKFRV), which contains 26 AA residues, was synthesized by a solid-phase peptide synthesis technique (AnyGen Co., Gwangju, Korea), and purified by reverse-phase high-performance liquid chromatography using a SHIMADZU C18 analytical column (Shimadzu HPLC LabSolution, Kyoto, Japan). Mass analysis was conducted using linear MALDI-TOF mass spectrometry (AXIMA Assurance, MALDI-TOF; Shimadzu, Kyoto, Japan). The three-dimensional structure of octoprohibitin was generated using APPTEST (https://research.timmons.eu/apptest, accessed on 7 January 2022), and the images were visualized using Discovery Studio Visualizer (Version 21.1.0.20298, Biovia, CA, USA). The helical wheel projection of octoprohibitin was derived by Netwheel, Peptides Helical Wheel, and Net projection maker (http://lbqp.unb.br/etWheels/, accessed on 7 January 2022) [18].

### 4.2. Analysis of Octoprohibitin Antibacterial Activity against A. baumannii

We conducted an initial screening of octoprohibitin activity against *A. baumannii* at pH 7.4 using the broth microdilution method. Since it showed relatively lower activity, we changed the pH of the bacterial culture broth to 7, 8, and 9 and conducted the sensitivity test similarly, and found that pH 8 was the most effective pH level. All experiments were conducted in LB broth and LB agar at pH 8. The antibacterial activity of octoprohibitin was determined by time-kill kinetics, MIC, MBC, and bacterial viability using standard assays.

MIC and time-kill kinetics assays were conducted using the broth microdilution method, whereas MBC was determined using the agar plating method according to the CLSI guidelines (M07-A10). Broth microdilution was conducted on *A. baumannii* (1 × 10^6^ colony-forming units per milliliter; CFU/mL) in a 96-well plate (200 µL/well), and octoprohibitin was treated with different concentrations (0–500 µg/mL). The plate was incubated at 25 °C for 24 h, and bacterial growth was evaluated at 3 h intervals by measuring the optical density at 595 nm (OD_595_) using a microplate spectrophotometer (Bio-Rad, Saint Louis, USA). Growth kinetics are illustrated for each treatment level in a line graph with OD_595_ vs. time. The lowest octoprohibitin concentration that did not cause any OD_595_ changes for 24 h (no bacterial growth) was selected as the MIC. To determine the MBC, octoprohibitin treated at ≥MIC bacterial broths (100 µL) from 96-well plates was spread on LB agar plates and incubated at 25 °C for 24 h. The lowest octoprohibitin concentration that did not produce colonies on the agar plate was selected as the MBC against *A. baumannii.* The same procedure was conducted using chloramphenicol (100 µg/mL) as a positive control to determine MIC and MBC.

To determine the reduction in bacterial viability with octoprohibitin treatment, the MTT assay was conducted according to the method described by Dananjaya et al. [46] Briefly, *A. baumannii* (1 × 10^6^ CFU/mL) was treated with octoprohibitin (0–400 µg/mL) and incubated at 25 °C for 4 h. Bacteria were isolated by centrifugation at 1500× *g* for 10 min and washed with 1 × PBS. The bacterial cells were then treated with 10 µL of MTT reagent (5 µg/mL) (Sigma Aldrich, St. Louis, MO, USA). After incubation for 30 min, 50 µL of dimethyl sulfoxide (DMSO) was added to the mixture (Sigma Aldrich, St. Louis City, USA) to dissolve the formazan, and the absorbance was measured using a microplate spectrophotometer at OD_595__._

### 4.3. FE-SEM Analysis for Morphological and Structural Changes in Octoprohibitin-Treated A. baumannii

To determine whether octoprohibitin induces morphological and ultrastructural alterations on *A. baumannii* surfaces, FE-SEM analysis was conducted as described by Jayathilaka et al. [13]. In brief, *A. baumannii* (1 × 10^6^ CFU/mL) was treated with the MIC (200 µg/mL) and MBC (400 µg/mL) of octoprohibitin, 1 × PBS as the negative control, and chloramphenicol (100 µg/mL) as the positive control. After incubation at 25 °C for 9 h, the bacterial cells were collected by centrifugation at 1500 × *g* for 10 min and washed with 1 × PBS. The isolated bacterial cells were pre-fixed with 2.5% glutaraldehyde (Pharmacia, Uppsala, Sweden) for 20 min. Each sample was washed with 1 × PBS and dehydrated using a series of ethanol concentrations (30, 50, 70, 80, 90, and 100%). Cells were coated with platinum using an ion sputter (E_1030, Hitachi, Japan) and observed under FE-SEM (MERLIN™, Carl Zeiss, Germany).

### 4.4. Analysis of Membrane Permeability Alterations and ROS Generation

Octoprohibitin-treated *A. baumannii* was subjected to PI uptake assay and H_2_DCFDA staining to determine permeability alterations and ROS generation, respectively. FCM was conducted to determine the fluorescence level, and CFMC was conducted to capture stained cell images, according to a previously described method [17,46]. Briefly, *A. baumannii* (1 × 10^6^ CFU/mL) was treated with octoprohibitin at MIC (200 µg/mL) and MBC (400 µg/mL), 1 × PBS was used as the negative control, and chloramphenicol (100 µg/mL) was used as the positive control. Bacterial cells were collected after incubation at 25 °C for 3 and 9 h for FCM and CFMC, respectively. Isolated cells were washed with 1 × PBS and resuspended in 1 mL of PBS.

For PI uptake assay, cells were treated with 50 µg/mL PI (Sigma Aldrich, Saint Louis, USA) and 40 µg/mL FDA (Sigma Aldrich, Saint Louis, USA) at room temperature (26 ± 2 °C) for 30 min in the dark. Similarly, samples for ROS generation were stained with H_2_DCFDA (50 μg/mL) (Invitrogen, Waltham, MA, USA) at room temperature for 30 min in the dark. Excess staining was removed by centrifugation and 1 × PBS washing.

To determine the fluorescence level, cell samples were resuspended in 1 mL of 1 × PBS, and the intensity was observed using flow cytometry (FACS Calibur, Becton Dickinson, Franklin Lakes, NJ, USA). For the PI uptake assay, the red fluorescence level was measured, and the green fluorescence level was measured for ROS generation.

To capture micrographs of fluorescent cells, bacterial samples were resuspended in 50 µL of 1 × PBS, and 5 µL was placed on a glass slide. Fluorescent cells were observed under CFMC with a scan head integrated into an Axiovert 200 M inverted microscope (Carl Zeiss, Jena, Germany). The red fluorescence of PI was observed at excitation and emission wavelengths of 535 and 617 nm, respectively. Green fluorescence of FDA and H_2_DCFDA was observed at the excitation and emission wavelengths of 488 and 535 nm, respectively.

### 4.5. Agarose Gel Retardation Assay for DNA Binding Affinity

Octoprohibitin-treated genomic DNA of *A. baumannii* was subjected to an agarose gel retardation assay to determine the DNA-binding ability according to the method described by Jayathilaka et al. [18]. Genomic DNA was isolated from *A. baumannii* according to the manufacturer’s protocol using an AccuPrep^®^ Genomic DNA Extraction Kit (Bioneer, Daejeon, South Korea). Isolated DNA was quantified using a NanoDrop One^C^ Microvolume UV–Vis spectrophotometer (Thermo Scientific™, Waltham, MA, USA). Octoprohibitin treatment was performed on 200 ng of genomic DNA at increasing ratios (0:1, 0.25:1, 0.5:1, 1:1, 2.5:1, 5:1, 7.5:1, and 10:1) and incubated at 37 °C for 4 h. The total amount of DNA was loaded onto a 1% agarose gel with 2 µL of loading buffer (Takara Bio Inc. Kusatsu, Shiga, Japan) and electroporated in a mini-submarine electrophoresis system (Mupid^®^-2plus, Takara Bio Inc. Kusatsu, Shiga, Japan) for 40 min. A photograph of the gel was captured using a gel imaging system (MaXidoc^Tm^ G2, DAIHAN^®^, Republic of Korea). The intensity of each gel band was quantified using ImageJ software (ImageJ, ver. 1.6, Rasband, WS, USA), and normalized to the intensity of the untreated gel band.

### 4.6. Biofilm Inhibition and Eradication Activity Assay of Octoprohibitin

Biofilm formation inhibition and eradication assays were conducted on *A. baumannii* planktonic and its biofilm, respectively, with CV staining according to the method described by Kim et al. [47]. *A. baumannii* was cultured in LB broth (pH 8) supplemented with 0.1% glucose to enhance and stabilize biofilm formation. For the biofilm inhibition assay, *A. baumannii* (1 × 10^6^ CFU/mL) broth was added to a 96-well plate (100 µL/well), treated with octoprohibitin at different concentrations (0–1000 µg/mL) and chloramphenicol (100 µg/mL) as a positive control, and incubated for 24 h. For the biofilm eradication assay, initially, *A. baumannii* (1 × 10^6^ CFU/mL) broth (100 µL/well) was added and allowed to form biofilms on the wall of the 96-well plate. After biofilm formation with 24 h incubation at 25 °C, the supernatant was removed and replaced with fresh media. Octoprohibitin (0–800 µg/mL) or chloramphenicol (100 µg/mL) was added to the biofilm and incubated for 24 h at 25 °C.

After octoprohibitin treatment, the biofilm formation inhibition and biofilm eradication assay plates were used for CV staining. The supernatant was removed, and the wells were washed with 1 × PBS to remove planktonic bacteria. Absolute methanol (100 µL/well) was used to fix the biofilms for 10 min. After removal of methanol, 100 µL of 0.1% CV (Sigma-Aldrich, Burlington, MA, USA) was used to stain the biofilm at room temperature for 30 min. The CV was then removed, and the wells were washed with 1 × PBS to remove unbound CV. Biofilm-bound CV was dissolved by adding absolute ethanol (100 µL/well) and mixing. The absorbance of each well was measured using a microplate spectrophotometer at OD_595_. The biofilm formation inhibition and eradication percentages were calculated. Biofilm formation inhibition/eradication % = [1 − (Ab _test/_Ab _negative control_)] × 100%, where Ab _test_ represents the absorbance value of octoprohibitin or chloramphenicol; Ab _negative control_ represents the absorbance of the negative control (PBS). Ninety percent of the biofilm formation inhibition concentration and eradication concentration were calculated as MBIC and MBEC, respectively.

### 4.7. Hemolysis Assay and Cytotoxicity Assay for Determination of Octoprohibitin Toxicity

To determine the toxicity level of octoprohibitin, a hemolysis assay on mouse RBCs and a cytotoxicity test on HEK293T cells (American Type Culture Collection ATCC-11268) were conducted. For the hemolysis assay, RBCs were collected by centrifuging mouse blood at 1500× *g* and washed with 1 × PBS. Then, RBCs were suspended in 1 × PBS and treated with octoprohibitin at different concentrations (0–800 µg/mL), with 1 × PBS as the negative control and 1% (w/v) Triton x-100 (Sigma Aldrich, Saint Louis, MO, USA) as the positive control. After incubation at 37 °C for 4 h, the supernatant was separated by centrifugation at 1500× *g* and the absorbance was measured using a microplate spectrophotometer at 490 nm. The hemolysis percentage was calculated by normalizing with the 100% hemolysis absorbance using 1% Triton X-100. Hemolysis (%) = ((Ab _test_ − Ab _PBS_)/(Ab _triton_− Ab _PBS_)) × 100%, where Ab _test_ represents the absorbance of octoprohibitin-treated RBCs supernatant, Ab _PBS_ represents the absorbance of PBS-treated RBCs supernatant, and Ab _triton_ represents the absorbance of Triton X-100-treated RBCs supernatant.

An MTT assay was conducted to determine the change in viability of HEK293T cells treated with octoprohibitin. Briefly, HEK293T cells were seeded in Dulbecco’s modified Eagle’s medium (Invitrogen, Carlsbad, CA, USA) with 1% antibiotic/antimycotic solution (Glibco, Carlsbad, CA, USA) and 10% fetal bovine serum (Hyclone, Carlsbad, CA, USA). Then, the cells were seeded in a 96-well plate at a cell density of 1.5 × 10^6^ cells/well and incubated at 37 °C in a 5% CO_2_ atmosphere. After 24 h of incubation, the supernatant was removed and replaced with a fresh medium. Octoprohibitin treatment was conducted in a concentration series (0–800 µg/mL) and incubated at 37 °C in a 5% CO_2_ atmosphere for 24 h. Culture media was replaced with 90 µL fresh media and 10 µL of MTT reagent (5 µg/mL). After incubation at 37 °C for 4 h, 50 µL DMSO was added to each well and mixed. Solubilized formazan was quantified by absorbance measurement using a microplate spectrophotometer at OD_595_.

### 4.8. In Vivo Octoprohibitin Efficacy Assay against A. baumannii-Infected Zebrafish

In vivo zebrafish experiments were performed according to the institutional animal ethics guidelines and under the supervision of the Committee of Chungnam National University (202103-CNU-072). Bacteria (*A. baumannii*) challenge and octoprohibitin treatment were conducted intraperitoneally in zebrafish according to the method described by Liyanage et al. [48]. Briefly, zebrafish were divided into three groups (30 fish per group in triplicate) and labeled; 1) only PBS-treated (negative control), 2) *A. baumannii* challenged and 1 × PBS-treated, and 3) *A. baumannii* challenged and octoprohibitin-treated. First, zebrafish were anesthetized using 160 μg/mL buffered tricaine (ethyl 3-aminobenzoate methanesulfonate; Sigma-Aldrich, USA) and injected with an infective dose of *A. baumannii* (20 µL of 2.1 × 10^11^ CFU/mL). Then, treatment was conducted with either 10 µL of octoprohibitin (10 µg/mL) or 10 µL of 1 × PBS. The negative control group was injected with 30 µL 1 × PBS. Fish were placed in tanks and maintained at 28 °C while observing mortality every 6 h for 48 h. The pathological signs were monitored in the remaining fish at 48 hpt. Finally, CS% was calculated.

For histopathological analysis, three zebrafish per group were sacrificed for collection of gut, kidney, and gill tissues at the end of the experiment. Histological analysis was performed according to the method described by Liyanage et al. [48]. The collected tissues were processed, embedded, and sectioned to a thickness of 4 µm, and glass slides were prepared for each section. Hematoxylin and eosin staining was performed, and the slides were observed under a light microscope (Leica 3000 LED, Wetzlar, Germany).

## 5. Conclusions

Octoprohibitin exhibited bactericidal activity against *A. baumannii* in a concentration-dependent manner, with MIC and MBC values of 200 and 400 µg/mL, respectively. *A. baumannii* growth and viability reductions were observed with increasing concentrations of octoprohibitin. Morphological and structural changes in the bacterial surface, membrane permeability alteration, ROS generation, and DNA-binding activity were determined as possible modes of action for the efficient killing activity of *A. baumannii* and preventing the development of antimicrobial resistance. In addition to its antimicrobial activity, octoprohibitin showed concentration-dependent antibiofilm activity, with biofilm formation inhibition and biofilm eradication. The MBIC and MBEC were 1000 and 1460 µg/mL, respectively. Hemolysis and cell viability assays confirmed that the toxic concentrations of octoprohibitin are above the MIC and MBC and are safer to use in vivo experiments. Finally, we found the efficiency of octoprohibitin in controlling *A. baumannii* not only in vitro but also in vivo, with higher survival of zebrafish and least damage to the gut, kidney, and gill tissues in the zebrafish model. Further experiments can descriptively explain the actions, safety, and application of octoprohibitin, and finally, introduce it as a pharmaceutical product to control multidrug resistance in nosocomial *A. baumannii.*

## Figures and Tables

**Figure 1 pharmaceuticals-15-00928-f001:**
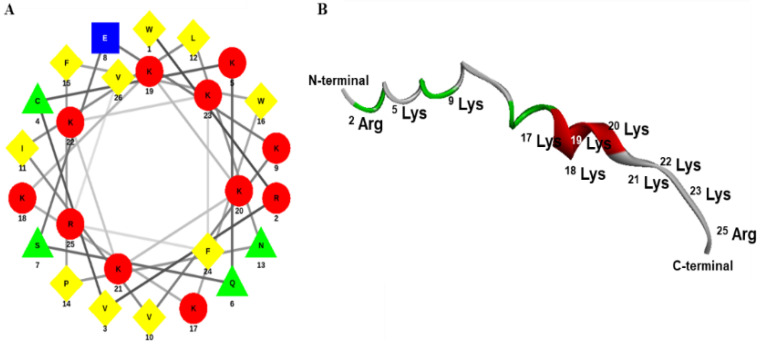
Octoprohibitin predicted secondary and tertiary structures. (**A**) Octoprohibitin helical wheel illustration. The AAs are arranged with the number of residues that are counted from the amino (N-) terminal. Basic residues and acidic residues are given in red colour circles and blue colour square, respectively, while polar and non-polar residues are demonstrated in a green colour triangle and yellow colour rhombus, respectively. (**B**) The tertiary structure (3D structure) of Octoprohibitin with positively charged residues are marked.

**Figure 2 pharmaceuticals-15-00928-f002:**
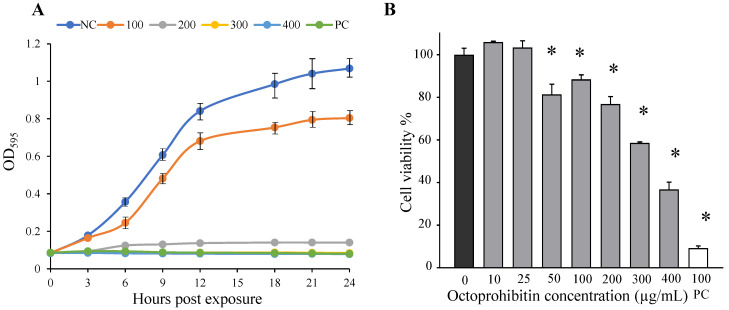
Octoprohibitin antibacterial activity against *A. baumannii*. (**A**) Time-kill kinetic of *A. baumannii*. Bacteria culture (1 × 10^6^ CFU/mL) was treated with octoprohibitin (0, 100, 200, 300, and 400 µg/mL) and bacterial growth was observed at every 3 h time intervals at OD_595_. As negative and positive controls, 1 × phosphate buffer saline (1 × PBS) and chloramphenicol (100 µg/mL), respectively, were used. The error bars indicate the mean ± standard deviation (*n* = 3). NC = Negative control, PC = positive control. (**B**) *A. baumannii* bacterial viability percentage with octoprohibitin treatment. Bacteria culture (1 × 10^6^ CFU/mL) was treated with octoprohibitin at different concentrations (0, 10, 25, 50, 100, 200, 300, and 400 µg/mL) and chloramphenicol as positive control. MTT assay was conducted on each sample after incubation for 24 h. An unpaired two-tailed *t*-test was conducted to derive the significant differences between the negative control and the treated samples. ** p* < 0.05 compared to the control (0) group. The error bars indicate the mean ± standard deviation (*n* = 3). PC = positive control.

**Figure 3 pharmaceuticals-15-00928-f003:**
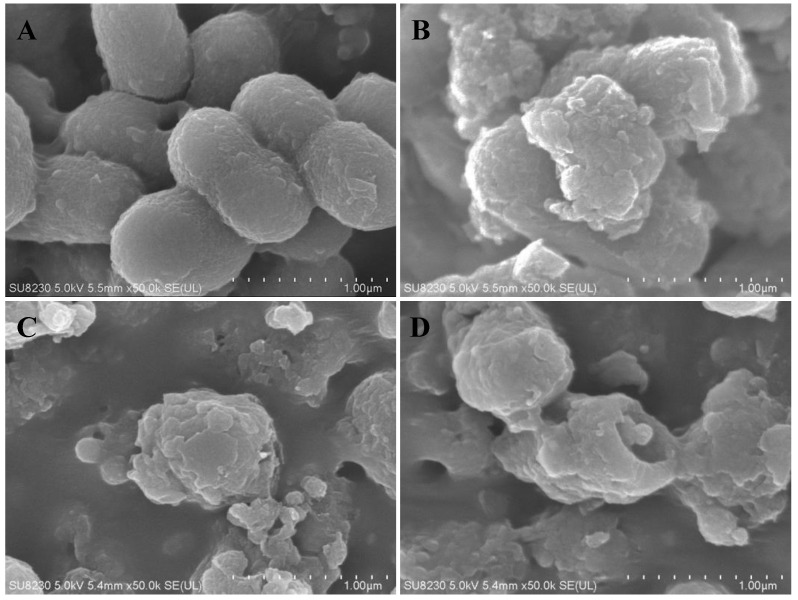
Morphological and structural changes of *A. baumannii* with octoprohibitin treatment. FE-SEM analysis of *A. baumannii* which were treated for 10 h with (**A**) PBS as a negative control, (**B**) MIC (200 µg/mL), (**C**) MBC (400 µg/mL) of octoprohibitin, and (**D**) chloramphenicol (100 µg/mL) as a positive control. Significant morphological and structural damages on *A. baumannii* cell surface with octoprohibitin treatment at MIC and total cell disruption at MBC were observed.

**Figure 4 pharmaceuticals-15-00928-f004:**
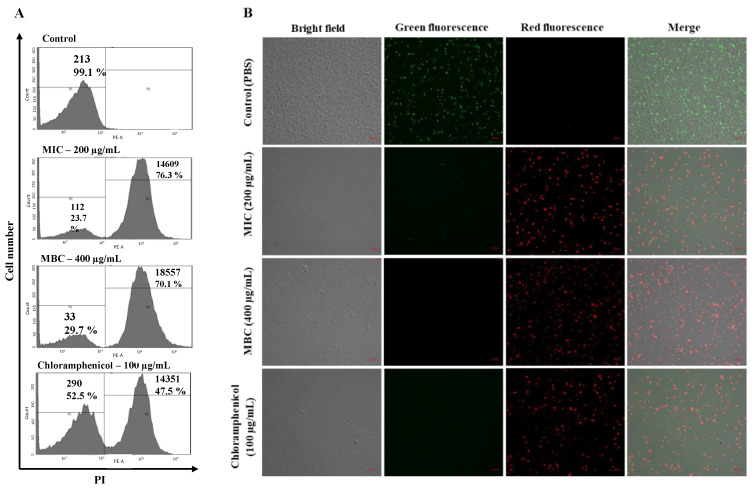
*A. baumannii* membrane permeability alteration with octoprohibitin treatments. (**A**) Flow cytometry analysis and (**B**) CLSM analysis on *A. baumannii* with octoprohibitin treatments at MIC (200 µg/mL), MBC (400 µg/mL), and chloramphenicol (100 µg/mL) as the positive control. Negative control was treated with 1 × PBS. After 10 h incubation, bacteria cells were stained with PI for membrane permeability altered cells and conducted flow cytometry analysis. PI- and FDA-stained cells were observed by CLSM and captured micrographs for red fluorescence (excitation—535 nm and emission—617 nm) and green fluorescence (excitation—488 nm and emission—535 nm). Scale bar; 10 μm.

**Figure 5 pharmaceuticals-15-00928-f005:**
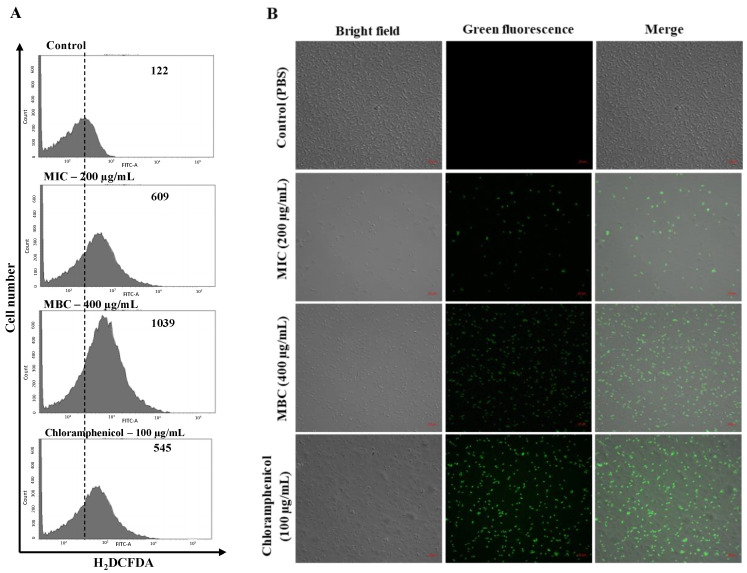
ROS generation of *A. baumannii* with octoprohibitin treatment. (**A**) Flow cytometry analysis for the fluorescence level intensity determination and (**B**) CLSM images of green fluorescence-stained cells for ROS production in *A. baumannii* at MIC (200 µg/mL) and MBC (400 µg/mL) treatments of octoprohibitin compared to that of the negative control (1 × PBS treated) and the positive control (chloramphenicol; 100 µg/mL treated). Bacteria cells at different treatments were stained with H_2_DCFDA and observed at an excitation and emission wavelength of 488 and 535 nm, respectively.

**Figure 6 pharmaceuticals-15-00928-f006:**
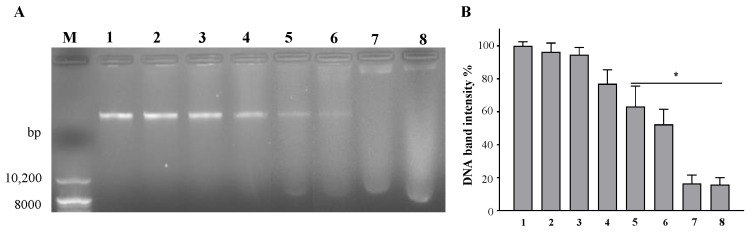
Agarose gel retardation assay of octoprohibitin-treated *A. baumanii* genomic DNA. Genomic DNA was reacted with octoprohibitin at different ratios ((1) 0:1, (2) 0.25:1, (3) 0.5:1, (4) 1:1, (5) 2.5:1, (6) 5:1, (7) 7.5:1, and (8) 10:1) and incubated at 37 °C for 4 h. (**A**) Total sample mixture was electroporated in 1% agarose gel. The binding ability of octoprohibitin with genomic DNA was assessed by the migration pattern of DNA. At 7.5:1 and above ratios of octoprohibitin, DNA migration was inhibited and smear formation was observed in higher concentrations of octoprohibitin. (M) DNA marker. (**B**) Quantification of DNA band intensities compared to that of the non-treated DNA sample band. An unpaired two-tailed *t*-test was conducted to derive the significant differences between the negative control and the treated samples. ** p* < 0.05 compared to the control (0) group. The error bars indicate the mean ± standard deviation (*n* = 3).

**Figure 7 pharmaceuticals-15-00928-f007:**
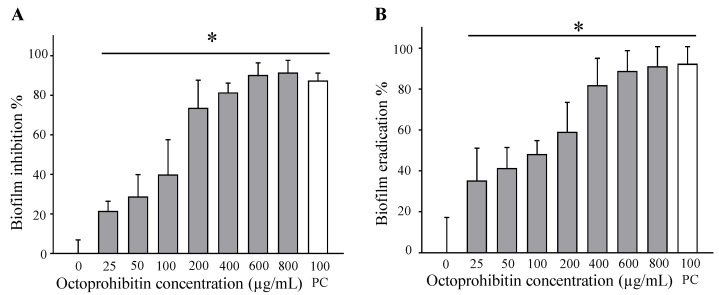
Antibiofilm effect of octoprohibitin against *A. baumannii* biofilm. (**A**) Biofilm formation inhibition with octoprohibitin treatment on *A. baumannii*. The bacterial culture was treated with octoprohibitin (0–800 µg/mL) and incubated for 24 h to form the biofilm. The remained biofilm after 24 h was stained by CV and quantified by absorbance measurement at OD_595_ (**B**) Biofilm eradication capacity of octoprohibitin on pre-formed *A. baumannii* biofilm. Bacteria were allowed to grow for the formation of the biofilm and treated with octoprohibitin (0–800 µg/mL). After 24 h incubation, remained biofilm was stained with CV and quantified by absorbance at OD_595_. Data are presented as the mean ± standard error (*n* = 3). An unpaired two-tailed *t*-test was conducted to derive the significant differences between the negative control and the treated (25–800 μg/mL) groups. ** p* < 0.05.

**Figure 8 pharmaceuticals-15-00928-f008:**
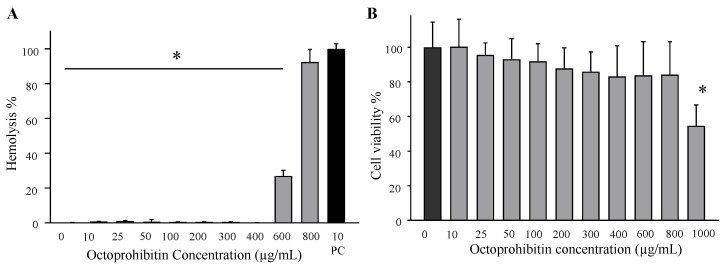
Hematotoxicity and cytotoxicity of octoprohibitin. (**A**) Hemolytic activity with octoprohibitin treatment on mouse RBCs. Washed RBCs were treated with octoprohibitin (0–800 µg/mL), 1 × PBS as the negative control, and Triton-X (10 µg/mL) as the positive control for 100% hemolysis. After 1 h incubation, RBCs were centrifuged, and the supernatant was separated for absorbance measurement at 490 nm. No hemolysis was observed up to 400 µg/mL of octoprohibitin. ** p* < 0.05 compared to the positive control (Triton-X 10 µg/mL) group. Bars indicate the mean ± standard deviation (*n* = 3). (**B**) MTT assay on octoprohibitin (0–1000 µg/mL)-treated HEK 293T cells (2.0 × 10^5^ cells/mL). No significant cell viability changes were observed in cells treated with octoprohibitin up to 800 µg/mL, ** p* < 0.05 compared to the negative control group. The error bars indicate the mean ± standard deviation (*n* = 3).

**Figure 9 pharmaceuticals-15-00928-f009:**
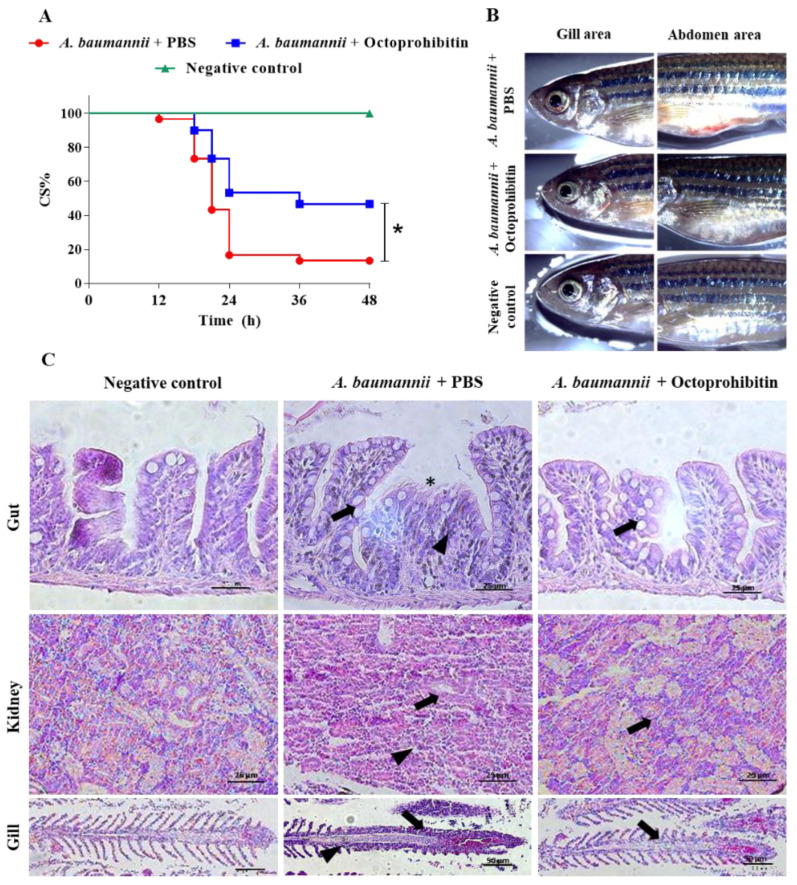
In vivo efficacy of octoprohibitin for controlling *A. baumannii* infection in the adult zebrafish model. (**A**) CS% of the adult zebrafish for 48 hpt. Zebrafish were challenged with *A. baumannii* intraperitoneally and treated with octoprohibitin or PBS. Negative control group was injected with PBS without *A. baumannii* challenge (**B**) Pathological signs of survived fish at 48 hpt in gill and abdomen area. (**C**) Histopathological analysis of gut, kidney, and gill tissues of negative control, *A. baumannii*-infected and PBS-treated and octoprohibitin treatment groups. Hyperplasia of goblet cells in gut tissue was shown in black arrow, erythrocyte infiltrations were shown in arrowhead, and epithelial damages were shown in asterisk. In kidney tissue, tubule infiltrations were shown in black arrow and tissue necrosis with low cell density was illustrated in arrowhead. In gill tissue, clubbing and shortening of secondary lamella were displayed in black arrow and thickening of primary lamella epithelial cells was shown in arrowhead.

## Data Availability

Data is contained within the article and Appendix A.

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
