# Peer review of "Novel Antimicrobial Peptide “Octoprohibitin” against Multidrug Resistant Acinetobacter baumannii"

_pharmaceuticals, 2022, doi:10.3390/ph15080928_

Round 1

Reviewer 1 Report

The manuscript by Jayathilaka et al. entitled “Novel antimicrobial peptide “Octoprohibitin” to challenge antibiotic resistance in Acinetobacter baumannii” reported a novel antimicrobial peptide (AMP), Octoprohibitin is a potential antibacterial and antibiofilm agent against MDR A. baumannii. The study looks interesting and well-conducted; however, the following comments should be addressed before taking a final decision on this manuscript.

1.          The abstract is too long of 319 words, and I suggest the authors follow the journal guidelines.

2.          Why did the authors select “Acinetobacter baumannii” as an objective pathogen, even though there are several MDR bacterial strains?

3.          I suggest adding the following articles in the introduction part.

https://www.nature.com/articles/nrmicro2693, https://doi.org/10.2147/IJN.S265934,  https://doi.org/10.3390/pharmaceutics14030582,

4.          The quality of figures 4A and 5A is poor. Please improve these figures as it is difficult to understand the text written in the figures.

5.          Improve the discussion part by comparing the results with recently published articles.

6.          Why did the authors use chloramphenicol as a negative control?

7.          What do the authors conclude that the novel AMP is bacteriostatic or bactericidal? Additionally, it may be effective against which type of bacteria?

8.          How do the authors check whether “Acinetobacter baumannii” is MDR or not? Furthermore, why the authors did not use the antibiotic-resistant (ABR) bacteria?

9.          Revise the conclusion as it looks not conclude all the important points.

Author Response

Manuscript ID - pharmaceuticals-1785196

Reviewer Report (Reviewer 1)

Title “Octoprohibitin” Novel antimicrobial peptide “Octoprohibitin” to challenge antibiotic resistance in Acinetobacter baumannii 

New title: Novel antimicrobial peptide “Octoprohibitin” against multidrug resistant Acinetobacter baumannii 

The manuscript by Jayathilaka et al. entitled “Novel antimicrobial peptide “Octoprohibitin” to challenge antibiotic resistance in Acinetobacter baumannii” reported a novel antimicrobial peptide (AMP), Octoprohibitin is a potential antibacterial and antibiofilm agent against MDR A. baumannii. The study looks interesting and well-conducted; however, the following comments should be addressed before taking a final decision on this manuscript.

Specific comments are as follows:

  1. Reviewer comments

The abstract is too long of 319 words, and I suggest the authors follow the journal guidelines.

Author’s comment

As suggested abstract was shortened (word count 239).

  1. Reviewer comments

Why did the authors select “Acinetobacter baumannii” as an objective pathogen, even though there are several MDR bacterial strains?

Author’s comment

In our previous study, we confirmed that A. baumanni has developed resistant to 11 antibiotics and intermediate resistance to 4 antibiotics (Jayathilaka et al., 2021). In a long run, our goal is to develop efficient AMPs to control resistant bacteria such as A. baumanni by applying AMPs delivery system. Therefore, we use A. baumanni as an drug resistant bacteria model which is convenient to make comparative study with different AMPs or its derivatives. 

Jayathilaka, E. T., Rajapaksha, D. C., Nikapitiya, C., De Zoysa, M., & Whang, I. (2021). Antimicrobial and anti-biofilm peptide octominin for controlling multidrug-resistant Acinetobacter baumannii. International Journal of Molecular Sciences, 22(10), 5353.

  1. Reviewer comments

I suggest adding the following articles in the introduction part.

https://www.nature.com/articles/nrmicro2693, https://doi.org/10.2147/IJN.S265934,  https://doi.org/10.3390/pharmaceutics14030582,

Author’s comment

The suggested references have been included in the revised manuscript.  (Ref. 8, 9, and 11).

  1. Reviewer comments

The quality of figures 4A and 5A is poor. Please improve these figures as it is difficult to understand the text written in the figures.

Author’s comment

Figures were rearranged for clear visualization of the labels.

  1. Reviewer comments

Improve the discussion part by comparing the results with recently published articles.

Author’s comment

We replaced several references by modifying the discussion with latest references.

Ref. 26, 32, 33, 37, 39, 40, and 41.

  1. Reviewer comments

Why did the authors use chloramphenicol as a negative control?

Author’s comment

In the antibiotic screening test, A. baumannii strain was susceptible to the chloramphenicol at low doses. We determined the MIC (50 µg/mL) and MBC (100 µg/mL) for chloramphenicol (Previous study; Jayathilaka et al., 2021). Upon that, we used chloramphenicol as a positive control to compare the antibacterial activity effect with Octoprohibitin.

Jayathilaka, E. T., Rajapaksha, D. C., Nikapitiya, C., De Zoysa, M., & Whang, I. (2021). Antimicrobial and anti-biofilm peptide octominin for controlling multidrug-resistant Acinetobacter baumannii. International Journal of Molecular Sciences, 22(10), 5353.

  1. Reviewer comments

What do the authors conclude that the novel AMP is bacteriostatic or bactericidal? Additionally, it may be effective against which type of bacteria?

Author’s comment

The MIC and MBC values of Octoprohibitin were 200 and 400 µg/mL, respectively. The ratio MBC/MIC is 2 (below than 4). Because of that, we conclude that Octoprohibitin produce bactericidal activity rather than bacteriostatic activity. Additionally, A. baumannii is Gram negative bacteria we suggest Octoprohibitin may be more effective against other Gram negative bacteria. However, our previous tested AMP was effective against both gram negative and positive bacteria. So further studies are essential for concluding exact which strains are susceptible for Octoprohibitin.

  1. Reviewer comments

How do the authors check whether “Acinetobacter baumannii” is MDR or not? Furthermore, why the authors did not use the antibiotic-resistant (ABR) bacteria?

Author’s comment

It was confirmed in our previous study (Jayathilake et al., 2021).

Jayathilaka, E. T., Rajapaksha, D. C., Nikapitiya, C., De Zoysa, M., & Whang, I. (2021). Antimicrobial and anti-biofilm peptide octominin for controlling multidrug-resistant Acinetobacter baumannii. International Journal of Molecular Sciences, 22(10), 5353.

  1. Reviewer comments

Revise the conclusion as it looks not conclude all the important points.

Author’s comment

As suggested conclusion was reorganized in the revised manuscript.  

Reviewer 2 Report

The manuscript „Novel antimicrobial peptide “Octoprohibitin” to challenge antibiotic resistance in Acinetobacter baumannii” by E. H. T. Thulshan Jayathilaka and co-workers was submitted to the journal “Pharmaceuticals” for consideration to be published as an “Article”.

The article deals with the characterization of the antimicrobial peptide Octoprohibitin, generally covering the antimicrobial activity against A. baumanii and in a zebrafish model. In particular, the authors performed studies to get insights into the mechanism of action (morphological changes, influence on membrane permeability, generation of reactive oxygen species, binding to DNA, effect on formation of biofilms.

The manuscript is very similar to previous work (https://doi.org/10.3390/md18010056 and https://doi.org/10.1016/j.fsi.2021.07.019) of the research group on Octominin and Octopromycin. Therefore, the current study can be understood as a continuation of the research pursued by the group. It generally matches the scope of the section “Pharmacology” of the journal “Pharmaceuticals”.

“…and we  have synthesized Octominin and Octopromycin in previous studies.” The authors are kindly asked to cite their previous work directly.

Can the authors provide information about the synthesis and analytical characterization of Octoprohibitin in the running text?

“…and found Octoprohibitin to show efficient antibacterial activity with increasing pH (Supplementary figure 1).” Unfortunately there was no supplementary material available in the submission/review portal.

Figure 2A: please indicate the blue cure as “NC” in accordance to “PC”. Are these data the mean ± standard deviation of n = ? independent experiments? Please show the error bar in the figure and provide a short comment on the way of data presentation in the caption of the figure.

Figure 2B: The asterisks have slipped (also Figure 8B). Please adjust. In addition, please provide in the caption which statistical test (probably t-test) was used to calculate statistical significance(also Figure 8B).

Figure 6B: Was this experiment performed just one time (since error bars are missing)?

Lines  283-286, 290: four times “challenged”. Please use synonyms to make your running text more appealing.

Line 310: CS% vs. RPS% in the figure?

Figure 9C: The caption should not provide a description or interpretation of the results. This should be done – in my opinion – in the running text.

“AMPs can be considered the next major solution for the control of MDR pathogens” and “and finally introduce it as a pharmaceutical product to control multidrug resistance, nosocomial A. baumannii.” In fact, AMPs have many advantages and, consequently, they also have multiple potentials to fight (resistant) bacteria. However, I would not currently say that they represent the "next major solution" to solve the problem of the therapy of resistant bacteria. In my opinion, the clinical situation does not yet allow to attribute such "winning" properties to AMPs. Therefore, it is recommended to tone down the direct statement somewhat.

Can the authors provide a comment/estimation on the potential of Octoprohibitin serving as antifungal compound and also on the possibility to serve as anticancer compound (e.g., formation of ROS, interaction with DNA).

Although there will be a close editing by the MDPI publisher, the authors are asked to correct formal errors / inconsistencies such as punctuation (lines 43, 159, 188/189, 291, 361, 440), use of space (186 vs. 208), typos (lines 238, 243: Figure 7A/B), use of bold (line 251), case shift (x-axis Figure 8A/B, 378) among others.

Author Response

Manuscript ID - pharmaceuticals-1785196

Reviewer Report (Reviewer 2)

Title “Octoprohibitin” Novel antimicrobial peptide “Octoprohibitin” to challenge antibiotic resistance in Acinetobacter baumannii 

New title: Novel antimicrobial peptide “Octoprohibitin” against multidrug resistant Acinetobacter baumannii 

Comments and Suggestions for Authors

The manuscript Novel antimicrobial peptide “Octoprohibitin” to challenge antibiotic resistance in Acinetobacter baumannii” by E. H. T. Thulshan Jayathilaka and co-workers was submitted to the journal “Pharmaceuticals” for consideration to be published as an “Article”.

The article deals with the characterization of the antimicrobial peptide Octoprohibitin, generally covering the antimicrobial activity against A. baumanii and in a zebrafish model. In particular, the authors performed studies to get insights into the mechanism of action (morphological changes, influence on membrane permeability, generation of reactive oxygen species, binding to DNA, effect on formation of biofilms.

The manuscript is very similar to previous work (https://doi.org/10.3390/md18010056 and https://doi.org/10.1016/j.fsi.2021.07.019) of the research group on Octominin and Octopromycin. Therefore, the current study can be understood as a continuation of the research pursued by the group. It generally matches the scope of the section “Pharmacology” of the journal “Pharmaceuticals”.

Specific comments are as follows:

  1. Reviewer comments

“…and we have synthesized Octominin and Octopromycin in previous studies.” The authors are kindly asked to cite their previous work directly.

Author’s comment

All relevant references have been cited.

  1. Reviewer comments
    Can the authors provide information about the synthesis and analytical characterization of Octoprohibitin in the running text?

Author’s comment

Octoprohibitin sequence was designed by us and it was synthesized at AnyGen Co., Gwangju, Korea. Company provided the HPLC analysis report and purity data. We submitted the HPLC graphs as supplementary figure 1.

  1. Reviewer comments

“…and found Octoprohibitin to show efficient antibacterial activity with increasing pH (Supplementary figure 1).” Unfortunately, there was no supplementary material available in the submission/review portal.

Author’s comment

Supplementary figure 2 has been included.

  1. Reviewer comments 4

Figure 2A: please indicate the blue cure as “NC” in accordance to “PC”. Are these data the mean ± standard deviation of n = ? independent experiments? Please show the error bar in the figure and provide a short comment on the way of data presentation in the caption of the figure.

Author’s comment

The figure labeling was corrected as suggested. And included error bar, n=3, and the data presentation.

  1. Reviewer comments

Figure 2B: The asterisks have slipped (also Figure 8B). Please adjust. In addition, please provide in the caption which statistical test (probably t-test) was used to calculate statistical significance (Figure 8B).

Author’s comment

We added the asterisks in figure 2B. However, in figure 8B the significance deviation was not observed in statistical analysis due to error margin. So the Figure 8B is placed without any corrections.

  1. Reviewer comments 6

Figure 6B: Was this experiment performed just one time (since error bars are missing)?

Author’s comment

The DNA retardation assay was conducted with replicates. However, we interpreted data in graph according to the data obtained the mentioned figure 6A. As you pointed out we used the other replicate data and conducted the statistical analysis also. Corrected graph with standard error and statistical data are replaced.

  1. Reviewer comments

Lines 283-286, 290: four times “challenged”. Please use synonyms to make your running text more appealing.

Author’s comment

We replaced the repetitive word with synonyms. Line 286- 289 and 293.

  1. Reviewer comments

“AMPs can be considered the next major solution for the control of MDR pathogens” and “and finally introduce it as a pharmaceutical product to control multidrug resistance, nosocomial A. baumannii.” In fact, AMPs have many advantages and, consequently, they also have multiple potentials to fight (resistant) bacteria. However, I would not currently say that they represent the "next major solution" to solve the problem of the therapy of resistant bacteria. In my opinion, the clinical situation does not yet allow to attribute such "winning" properties to AMPs. Therefore, it is recommended to tone down the direct statement somewhat.

Author’s comment

As suggested section was modified accordingly. Line 324.

  1. Reviewer comments

Can the authors provide a comment/estimation on the potential of Octoprohibitin serving as an antifungal compound and also on the possibility to serve as an anticancer compound (e.g., formation of ROS, interaction with DNA).

Author’s comment

In our previous study, Octominin (AMP) showed both antifungal activity and antibacterial activity against C. albicans and also A. baumannii. Characteristically, Octoprohibitin showed similar properties compared to the Octoprohibitin. Further Octoprohibitin produced similar modes of action so with further experiment we hope to derive Octoprohibitin’s antifungal activities also.

We included a separate paragraph for the manuscript discussing this point.  Line 455- 461. Even though ROS generation was observed in bacterial cells, we need to conduct for carcinoma cell (eukaryotic) lines to observe the ROS generation and DNA binding with Octoprohibitin to conclude it as an anticancer drug in future experiments.

  1. Reviewer comments

Although there will be close editing by the MDPI publisher, the authors are asked to correct formal errors/inconsistencies such as punctuation (lines 43, 159, 188/189, 291, 361, 440), use of space (186 vs. 208), typos (lines 238, 243: Figure 7A/B), use of bold (line 251), case shift (x-axis Figure 8A/B, 378) among others.

Punctuation corrections – Line – 41, 162, 190-191, 438

Use of space – 188, 210

Typos – 242, 247

Use bold – 255

Case lift – X axis Figure 8B

Author’s comment

We have obtained the English editing service for revised manuscript and certificate is attached.

Round 2

Reviewer 1 Report

I thank the authors that they have addressed the suggested queries. Therefore, in my opinion, the manuscript is now acceptable in its current form.